# The impact of short term Antiretroviral Therapy (ART) interruptions on longer term maternal health outcomes—A randomized clinical trial

**Patience Atuhaire**[1]*, **Sean S. Brummel**[2], **Blandina Theophil Mmbaga**[3], **Konstantia Angelidou**[4], **Lee Fairlie**[5], **Avy Violari**[6], **Gerhard Theron**[7], **Cornelius Mukuzunga**[8], **Sajeeda Mawlana**[9], **Mwangelwa Mubiana-Mbewe**[10], **Megeshinee Naidoo**[11], **Bonus Makanani**[12], **Patricia Mandima**[13], **Teacler Nematadzira**[14], **Nishi Suryavanshi**[15], **Tapiwa Mbengeranwa**[16], **Amy Loftis**[17], **Michael Basar**[18], **Katie McCarthy**[19], **Judith S. Currier**[20], **Mary Glenn Fowler**[21], **for the 1077BF/1077FF PROMISE Team**[¶]

1 Makerere University –John Hopkins University Research Collaboration (MUJHU CARE LTD) CRS, Kampala, Uganda, 2 Harvard School of Public Health, Boston, Massachusetts, United States of America, 3 Kilimanjaro Christian Medical Centre (KCMC), Moshi, Tanzania, 4 Harvard School of Public Health, Boston, Massachusetts, United States of America, 5 Wits RHI Shandukani Research Centre CRS, Johannesburg, South Africa, 6 Soweto IMPAACT CRS, Johannesburg, South Africa, 7 FAM-CRU CRS, Cape Town, South Africa, 8 Malawi CRS, Lilongwe, Malawi, 9 Durban Paediatric HIV CRS, Durban, South Africa, 10 George CRS, Lusaka, Zambia, 11 Umlazi CRS, Durban, South Africa, 12 Blantyre CRS, Blantyre, Malawi, 13 St Marys CRS, Harare, Zimbabwe, 14 Seke North CRS, Harare, Zimbabwe, 15 Byramjee Jeejeebhoy Medical College (BJMC) CRS, Pune, India, 16 Harare Family Care CRS, Harare, Zimbabwe, 17 UNC, Chapel Hill, North Carolina, United States of America, 18 Frontier Science, Amherst, Massachusetts, United States of America, 19 FHI 360, Durham, North Carolina, United States of America, 20 UCLA center for Clinical AIDS Research and Education, Los Angeles, California, United States of America, 21 Departments of Pathology and Epidemiology, Johns Hopkins University, Baltimore, Maryland, United States of America

¶ Membership of the 1077BF/1077FF PROMISE Team is provided in the Acknowledgments.
* patuhaire@mujhu.org

**Data Availability Statement:** Due to ethical restrictions in the study's informed consent documents and in the IMPAACT Network's

## Abstract

### Background

Given well documented challenges faced by pregnant women living with HIV taking lifetime ART, it is critical to understand the impact of short-term ART exposure followed by treatment interruption on maternal health outcomes.

### Methods

HIV+ breastfeeding (BF) and Formula Feeding (FF) women with CD4 counts > 350 cells/mm3, enrolled in the 1077BF/1077FF PROMISE trial were followed to assess the effect of ART during pregnancy and breastfeeding respectively. The first analysis compared ART use limited to the antepartum period (AP-only) relative to women randomized to Zidovudine. The second analysis included women with no pregnancy combination ART exposure; and compared women randomized to either ART or no ART during postpartum (PP-only). Both analyses included follow-up time beyond breastfeeding period. The primary outcome was

approved human subjects protection plan, study data are available upon request from sdac.data@sdac.harvard.edu with the written agreement of the International Maternal Pediatric Adolescent AIDS Clinical Trials (IMPAACT) network. Data are also available to all interested researchers upon request to the IMPAACT Statistical and Data Management Center's data access committee (email address: sdac.data@fstrf.org); this committee reviews and responds to requests for data, obtains necessary approvals from IMPAACT leadership and the NIH, arranges for signature of a Data Use Agreement, and sends the requested data.

**Funding:** Funding for this study was provided by the National Institute of Allergy and Infectious Diseases (NIAID) of the National Institutes of Health (NIH) under Award Numbers UM1AI068632 (IMPAACT LOC), UM1AI068616 (IMPAACT SDMC) and UM1AI106716 (IMPAACT LC), with co-funding from the Eunice Kennedy Shriver National Institute of Child Health and Human Development (NICHD) and the National Institute of Mental Health (NIMH). Study drugs were provided by AbbVie, Boehringer-Ingelheim, Gilead Sciences, and ViiV/GlaxoSmithKline.

**Competing interests:** The authors have declared that no competing interests exist.

progression to AIDS and/or death. Secondary outcomes included adverse events and HIV-related events.

## Results

3490 and 1137 HIV+ women were enrolled from 14 sites in Africa and India from April 2011 through September 2014 in cohort AP-only and PP-only, respectively. Most were Black African (96%); median age was 27 years; 97% were WHO Clinical Stage I; and most had a screening CD4 count $\geq$500 cells/mm3 (78%). The rate of progression to AIDS and/or death was similar and low across all comparison arms (AP comparison, HR = 1.14, 95%CI (0.44, 2.96), p-value = 0.79). In the PP-only cohort, the rate of WHO stage 2–3 events was lower for women randomized to ART(HR = 0.65, 95% CI 0.42, 1.01, p-value = 0.05).

## Conclusion

The incidence of AIDS and/or death was low in pregnant/postpartum HIV+ women with highCD4 cell counts for all comparison arms. This provides some reassurance that there were limited consequences for short term ART interruption in this group of asymptomatic HIV+ women during up to 4 years of follow up; and underscores that even short term ART exposure postpartum may reduce the risk of WHO grade 2–3 disease progression.

## Introduction

In 2018, the Joint United Nations Programme on HIV/AIDS (UNAIDS) reported that 18.8 million women and girls were living with HIV and most were of child bearing age[1]. In 2015 the World Health Organization (WHO) formally recommended the use of lifelong combination Antiretroviral Therapy (ART) initiated at the time of diagnosis [2]. This policy has now been adopted by most countries in resource limited settings including for pregnant and lactating mothers[3]

Many HIV infected women are first identified during routine antenatal rapid HIV testing using the "Test and Treat" approach recommended by WHO. However, adherence to taking life time ART has been particularly challenging for HIV infected mothers during pregnancy, childbirth and postpartum [4]. In addition, antiretrovirals (ARVs) have both short term and long term side effects/toxicities which also concern mothers and may lead to lapses in adherence [5, 6].

Given these concerns about maternal adherence to lifetime ART, it is critical to better understand the consequences of short term ART followed by periods of treatment interruption. The International Maternal Pediatric Adolescent AIDS Clinical Trials Group (IMPAACT) PROMISE 1077BF/1077 randomized trial conducted in resource limited settings in Africa and India, was specifically designed to address the impact of ART on maternal health, including safety and efficacy, among asymptomatic HIV infected women with high CD4 counts, who at the time of the trial, did not meet country criteria for ART. The sequential randomizations in PROMISE provided a unique opportunity to compare the effects on maternal health of short term ART given only during pregnancy when compared to other proven perinatal HIV transmission ARV regimens; and likewise of maternal ART given only post- partum, during breastfeeding.

Previous analyses of maternal health outcomes from the PROMISE trial have focused on comparisons of women who were randomized to stop or continue ART in the post-partum period [7, 8]. The pre-planned maternal health sub cohort analyses presented here extends the maternal follow-up time of these two prior analyses past the time of transmission risk during pregnancy and breastfeeding, and after cessation of breastfeeding in order to better understand the impact of short term ART interventions on longer term maternal health outcomes. Specifically, this analysis focuses on two pre-specified comparisons of the effect of maternal combination ART used only during the antepartum period (cohort AP-only); and the effect of maternal combination ART given only during the breastfeeding period (cohort PP-only) followed by treatment interruption unless the women met standard- of- care treatment criteria.

## Study design, materials and methods

The IMPAACT 1077BF/1077FF PROMISE study was an open label multi-site clinical trial conducted in resource-limited settings which included up to 3 sequential randomized components. The random allocation sequence was generated by the Study Enrollment System (SES) that is located on the Data Management Center (DMC) of IMPAACT's website. The random assignment was conducted by the SES using stratified permuted blocks. The clinical site staff did not have access to the full permuted block, thus, the site clinical staff were not able to predict the random sequence. A candidate was randomized when a clinic staff person from an institution used the enrollment screens developed by FSTRF, the Data Management Center (DMC) of IMPAACT. The enrollment screens were accessible via the Study Enrollment System (SES), located on the DMC web site. In order to enroll candidates, the clinic staff member at the institution must have been having a DMC web site password and enrollment privileges. The candidate's eligibility was checked by answering questions in the Subject Enrollment System (SES). Once all eligibility questions had been answered and passed, the Patient Registration System (PRS) read the record that was generated from the eligibility check in SES. For eligible candidates, PRS performed a stratified randomization of treatment (if applicable), registered the participant on the DMC's central computer, and gave the participant a SID (Study Identification Number). The SID was then given to the site pharmacist who compared it to the SID list containing treatment information for the particular study and he/she would dispense study drug.

The AP-only analysis used a block size of 4 for period 1 that had 2 randomization arms, and a block size of 6 for period 2 that had 3 randomization arms. Randomization for the AP-only analysis was stratified by HBV status and country. The PP-Only analysis used a block size of 4. The PP-Only analysis randomization was stratified by type of maternal ARV prophylaxis during the Antepartum Component or Late Presenters Component (triple ARV prophylaxis regimen vs. ZDV + sdNVP + TRV tail vs. ZDV + sdNVP + TRV tail (late presenter) vs. none (late presenter)) of PROMISE and by country.

The study compared the relative efficacy and safety of maternal ART compared to other proven perinatal HIV transmission antiretroviral strategies during pregnancy and during breastfeeding. The trial included a Maternal Health component that assessed the impact of continued ART on the mother's health after the risk of mother-to-child transmission had ceased. The 1077BF trial was conducted in 14 sites in 7 countries in East and Southern Africa (Zambia, Zimbabwe, Malawi, Uganda, S. Africa, Tanzania); and in Pune, India.

The detailed methods of the Antepartum, Postpartum, and Maternal Health components of PROMISE have already been published [6, 7, 9]. This pre-specified Maternal Health analysis uses the randomizations and follow-up time from the three components of PROMISE to form two separate comparisons. The first comparison (AP-only), among women who at the time

did not meet country guidelines for treatment, was designed to estimate the effect of ART that is attributable to ART received *only during the antepartum period*. Triple ARV (ART) regimen in this group was started during pregnancy at 14 weeks or later gestation and stopped (i.e. treatment interruption) at 7–14 days post-delivery at the time of the next PROMISE randomization. For this first comparison analyses, follow up then continued for those mothers who received triple ARVs during pregnancy but then no triple ARVs throughout the breastfeeding period unless the woman met criteria for treatment during follow up. The second comparison group (PP-only) was designed to estimate the effect of ART when taken o*nly during the post-partum period*. Triple ARV regimen in this group was initiated at 7–14 days post-delivery. The primary outcomes were progression to AIDS or death; and relative safety of ART compared to the other regimens used.

**Ethical considerations and interim monitoring.**   All women provided written informed consent. The study was approved by local and collaborating institutional review boards and other relevant regulatory authorities; and was reviewed for safety and efficacy by an independent Data and Safety Monitoring Board (DSMB). These include MUJHU/Kampala, Uganda: The Joint Clinical Research Centre (JCRC) IRB, the National Drug Authority in Uganda and the Johns Hopkins Medical Institutions (JHMI) IRB in the U.S.; Wits RHI Shandukani CRS and Soweto IMPAACT CRS, Johannesburg, South Africa: University of Witwatersrand Human Ethics Research Committee (Medical), Medicines Control Council (South African Health Products Regulatory Authority in February 2018); FAM-CRU CRS, Cape town, South Africa: Health Research Ethics Committee (HREC), Faculty of Health Sciences, Stellenbosch University and Medicines Control Council (South African Health Products Regulatory Authority in February 2018); Durban Paediatric HIV CRS, Durban, South Africa: University of KwaZulu-Natal (UKZN) Biomedical Research Ethics Committee, Medicines Control Council (South African Health Products Regulatory Authority in February 2018);George CRS, Lusaka, Zambia: University of North Carolina (UNC) at Chapel Hill Biomedical IRB and University of Zambia Biomedical Research Ethics Committee (UNZABREC); Harare, Seke North and St. Mary's sites, Zimbabwe: Medical Research Council of Zimbabwe(MRCZ), Research Council of Zimbabwe (RCZ), Medicine Control Authority of Zimbabwe(MCAZ), Joint Parirenyatwa group of Hospitals/University of Zimbabwe College of Health Sciences Research Ethics Committee(JREC); Byramjee Jeejeebhoy Medical College (BJMC) CRS, Pune, India: BJ Government College CTU Ethics Committee and Johns Hopkins IRB; Blantyre, Malawi: College of Medicine Research and Ethics Committee (COMREC) in Malawi, Pharmacy, Medicines and Poisons Board and Johns Hopkins Medical Institutions (JHMI) IRB in the U.S.; Lilongwe, Malawi: National Health Sciences Research Committee (NHSRC) in Malawi Pharmacy, Medicines and Poisons Board, and University of North Carolina, Chapel Hill (UNC-CH) Office of Human Research Ethics IRB in the U.S and Kilimanjaro Christian Medical Centre (KCMC), Moshi, Tanzania: Kilimanjaro Christian Medical College Ethics Committee, National Health Research Ethics Committee and Tanzania Medicines and Medical Devices Authority.

**Study population and eligibility criteria.**   The AP-only cohort analyses focused on disease progression among those participants who met the eligibility criteria for the antepartum component[6] and who received either ZDV (Zidovudine) +sdNVP(single dose nevirapine) +TRV (Tenofovir/emitricitabine)1 week "tail", or received maternal ART; and then received no ARVs post-partum. The PP-only cohort analysis included women who met the eligibility criteria for randomization into the postpartum component of PROMISE[7] and had been on ZDV+sdNVP+TRV tail during pregnancy and who then met post-partum eligibility criteria and were randomized to either maternal ART or no maternal ART (i.e. on the study arm whose infants received daily infant nevirapine (NVP)). Both groups included follow-up time beyond breastfeeding cessation.

**Study procedures.** All participants were followed for at least 96 weeks after the last delivery in the Antepartum Component. Participants were seen for clinical and safety evaluations during pregnancy every 4 weeks till delivery; at 3, 6, 14, 26, 38 and 50 weeks after delivery and then every 12 weeks thereafter until study end. CD4 counts were measured at each study visit. Plasma HIV-1 RNA was measured at each visit for women who were randomized to ART or who during the trial met criteria according to country standard of care, to start lifelong ART; whereas plasma was stored and then run in batch for women not on ART.

**Study outcomes.** The primary outcome was progression to AIDS or death. Secondary efficacy outcomes included time to WHO II/III clinical events. The secondary safety outcome included selected Grade 2 laboratory abnormalities (renal, hepatic and hematologic) and all Grade 3 or higher laboratory values and signs and symptoms. Events were graded using the DAIDS Table for Grading the Severity of Adult and Pediatric Adverse Events, 2004 Version 1.0 (clarification August 2009) [10].

**Follow-up time, randomization, and comparison groups.** The AP-only analysis group includes women who were randomized to one of three arms: ZDV+sdNVP+TRV "tail" (perinatal transmission strategy), or combination ART with either LPV/r+ZDV/3TC, or LPV+TDF/FTC during pregnancy. Analyses were summarized by the three randomized arms but efficacy analyses were compared statistically after combining the two ART groups while analyses of toxicity compared each of the randomized ART regimens to each other arm. The LPV+TDF/FTC arm was only available to hepatitis B co-infected women at the start of PROMISE (Period 1). A subsequent version of the protocol allowed randomization to LPV+TDF/FTC for all women regardless of hepatitis B status (Period 2), when further safety data on use of TDF during pregnancy had become available. Therefore the "by arm" safety analyses were restricted to the time in which there were contemporaneous randomizations. This analysis strategy follows the analysis strategy from the previously published PROMISE antepartum perinatal transmission study[6]. Follow-up included time from the antepartum period through breastfeeding and post-weaning (Subset Schema, Figs 1 and 2A). To estimate the longer term effects of ART, provided only during the AP period, women who were randomized to start or continue ART in a subsequent PROMISE component had their follow-up time censored at the time of the subsequent randomization.

The PP-only cohort included women without combination ART exposure in the AP period. These women were either randomized to ZDV+sdNVP+TRV tail in the PROMISE AP component or did not receive antenatal care due to late presentation into care. Women randomized to ART only after delivery were compared to women randomized to No maternal ART after delivery (infants received daily NVP throughout breastfeeding). The preferred PROMISE postpartum ART regimen was LPV+TDF/FTC. Follow-up included the time from the post-delivery randomization through breastfeeding and post-weaning (Figs 1 and 2B). To estimate the effect of ART specific to the postpartum period, women who were randomized to continue ART in a subsequent PROMISE component had their follow-up time censored at the time of the subsequent randomization.

Participants not initially randomized to ART were immediately started on ART if they met immunologic or clinical indications for treatment based on country standard-of-care. In addition, women randomized to ART were eligible to be switched to 2nd line ART regimens if they were intolerant or had immunologic treatment failure, had virologic failure after 6 months of treatment, or for other reasons based on their clinician's judgment. Immunologic failure was defined as confirmed decrease in CD4 count to less than any of the following: pre-ARV initiation level (i.e., the baseline CD4 count at study entry), or 50% of the participants peak levels, or 350 cells/mm3 or below the country-specific threshold for initiation of treatment, if that

| | PREGNANCY | BREASTFEEDING | POST WEANING |
|---|---|---|---|
| **COHORT AP (N=3543)** | Triple ART* (N=1994) | none | None |
| | ZDV+sd NVP+ TRV tail** (N=1549) | None | None |
| | AP-Only follow-up period; median 58 wks Maximum 130 wks | | |
| **COHORT PP (N=1137)** | ZDV+sd NVP+ TRV tail (N=1137) | Triple ART*** (N=572) | None |
| | | No triple ART(N=565) | None |
| | PP-Only follow-up period median 102wks Max 207 wks | | |

*AntepartumTriple ART regimens:  Zidovudine/lamivudine (ZDV/3TC)/Lopinavir/ritonavir (Lop/r)
   Or Tenofovir/emtricitabine (TDF/FTC)/Lopinavir/ritonavir (Lop/r
**Zidovudine (ZDV) + single dose NVP (sdNVP) to mothers at labor onset; followed by 1 week
   Truvada (TDF/FTC) "tail"
***Postpartum Triple ART regimen: Tenofovir/emtricitabine (TDF/FTC)/Lopinavir/ritonavir
(Lop/r)

**Fig 1. 1077BF/1077FF follow up and study comparison groups for AP-only and PP-only cohorts.**

threshold is > 350 cells/mm3. Virologic failure in the trial was defined as a described in an earlier publication [8].

Note: For both the AP-only and PP-only analysis groups, follow-up time was not censored when ART was started for a clinical indication.

**Statistical analysis.** The original sample size was determined by the number of women randomized to the relevant arms of the PROMISE 1077BF/FF Antepartum and Postpartum Components. It was anticipated that 4,400 women would participate in the AP-only analysis group and 2,592 women would participate in the PP-only analysis group. To calculate power, an annual event rate of 3.33% was assumed for women randomized to ART, and ~5% annual event rate was assumed for the non-ART comparison groups. Assuming a 5% annual loss-to-follow-up rate and accounting for the planned analysis censoring, there was 90% power for the AP-only and PP-only analysis groups, with a two-sided Type I error of 5%. Three thousand five hundred and forty three (81%) of the anticipated sample size included in the AP only analysis group while 1137(44%) were included in the PP only analysis group.

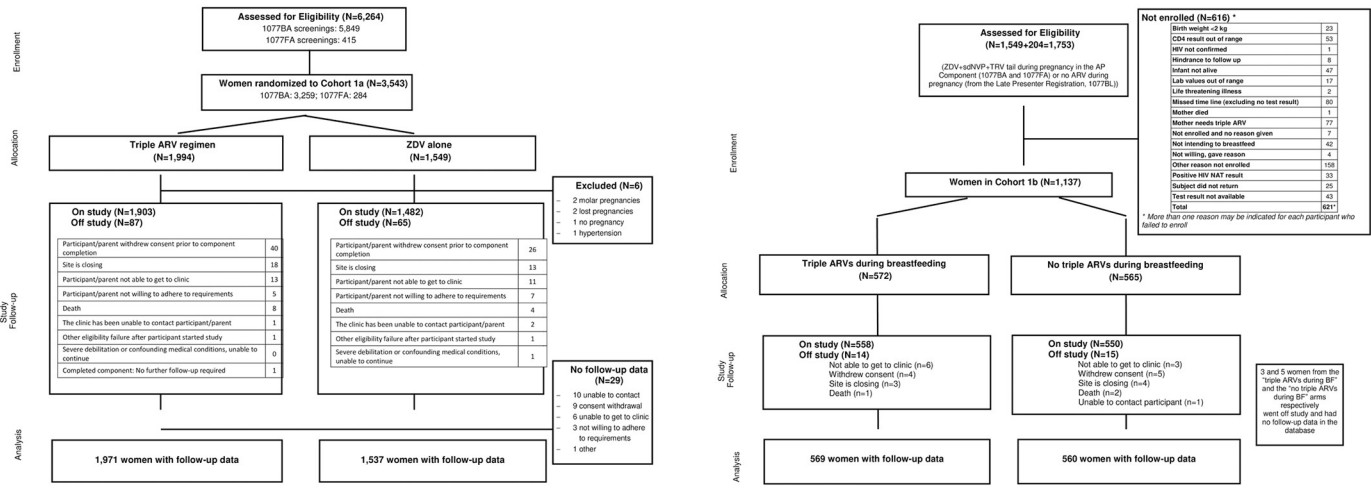

**Fig 2.** A. CONSORT diagram for the AP-only analysis group Fig 2B CONSORT diagram for the PP-only analysis group.

After the availability of the START trial results[2], demonstrating a benefit from ART regardless of CD4 cell count, on July 7, 2015 the PROMISE sites were notified to offer ART to all participants. Therefore PROMISE analyses only included follow-up through July 7, 2015, the randomized period of the trial.

Survival outcomes were summarized using Kaplan-Meier estimators, tested with the log-rank test, and Cox regression models were used to estimate treatment effect. Incidence rates are displayed per 100 person-years. A two-sided p-value less than 0.05 was considered statistically significant. Statistical analyses were performed using Statistical Analysis System (SAS) software version 9.4 (SAS Institute, Cary, NC).

## Results

### Baseline

The AP-only analysis group included 3543 women (Fig 1) with a median follow-up of 58 weeks (max: 130 weeks; person years: 4618). The PP-only analysis group included 1137 women (Fig 1) with a median follow-up of 102 weeks (max: 207 weeks; person years: 2279). Baseline characteristics were similar between the two analysis groups: most women were Black African (96%), with a median age of 27 years and BMI of 24.8 kg/m2. Most women were assessed as having WHO Clinical Stage I disease (97%) and the majority (78%) had a screening CD4 count $\geq$500 cells/mm3 (Table 1).

### AP-only cohort outcomes

**Starting life-long ART, changes in ART regimens, and virologic failure.** Seven hundred twenty one women (20.6%) of 3508 with follow up data were initiated on ART during follow up in the trial for their own health or if they required a regimen change. Of the women who needed ART for their own health, 548 (76%) were started due to immunologic failure; other reasons were due to changes in national guidelines for treatment initiation or progression to clinical WHO Stage III and TB. Eleven women switched their ART because of virologic failure, toxicities, or immunologic failure. By week 24 after enrollment, the incidence of virologic failure was 7% (95% CI: (6%, 10%)).

**Table 1. Baseline characteristics 1077BF/1077FF subgroup AP-only and PP-only cohorts.**

| Characteristic | | Randomization Arm antepartum (AP-Only) | | | Randomization Arm postpartum (PP-Only) | | |
|---|---|---|---|---|---|---|---|
| | | ZDV/3TC +LPV/r | TDF/FTC + LPV/r | ZDV+sdNVP +TDF/FTC | Triple ARVs | No Triple ARVs | Total |
| | | N = 1545 | N = 412 | N = 1547 | (N = 572) | (N = 565) | (N = 1,137) |
| Age at randomization [years] | N | 1545 | 412 | 1547 | 572 | **1,137** | 1,137 |
| | Min-Max | 18–44 | 18–40 | 18–50 | 18–44 | 18–47 | 18–47 |
| | Median (Q1-Q3) | 27 (23–30) | 26 (23–30) | 26 (23–30) | 27 (23–30) | 26 (23–30) | 27 (23–30) |
| Race | Asian (from Indian subcontinent) | 0 (0%) | 1 (0%) | 1 (0%) | 0 (0%) | 1 (0%) | 1 (0%) |
| | Black African | 1498 (97%) | 411 (100%) | 1499 (97%) | 548 (96%) | 541 (96%) | 1,089 (96%) |
| | Indian (Native of India) | 46 (3%) | 0 (0%) | 46 (3%) | 23 (4%) | 22 (4%) | 45 (4%) |
| | Colored | 1 (0%) | 0 (0%) | 1 (0%) | 1 (0%) | 1 (0%) | 2 (0%) |
| Weight [kg] | N | 1545 | 412 | 1547 | 572 | 565 | 1,137 |
| | Min-Max | 35–140 | 43–136 | 36–128 | 35–119 | 39–120 | 35–120 |
| | Median (Q1-Q3) | 65 (58–74) | 64 (58–75) | 64 (58–74) | 62 (55–71) | 61 (55–70) | 62 (55–70) |
| BMI [kg/m$^2$] | N | 1535 | 409 | 1545 | 571 | 565 | 1,136 |
| | Min-Max | 14.4–54.7 | 19.3–52.2 | 15.1–51.7 | 16.6–46.5 | 15.8–50.6 | 15.8–50.6 |
| | Median (Q1-Q3) | 26.3 (23.4–29.8) | 26.2 (23.5–29.9) | 25.9(23.5–29.5) | 24.8(22.3–28.1) | 24.7(22.2–8.0) | 24.8(22.3–28.1) |
| Number of live infants | 1 | 1440 (93%) | 381 (92%) | 1450 (94%) | 568 (99%) | 563 (100%) | 1,131 (99%) |
| | 2 | 19 (1%) | 8 (2%) | 23 (1%) | 4 (1%) | 2 (0%) | 6 (1%) |
| WHO clinical stage | Clinical stage I | 1506 (98%) | 405 (98%) | 1493 (97%) | 549 (96%) | 549 (97%) | 1,098 (97%) |
| | Clinical stage II | 34 (2%) | 7 (2%) | 50 (3%) | 22 (4%) | 15 (3%) | 37 (3%) |
| | Clinical stage III | 2 (0%) | 0 (0%) | 1 (0%) | 1 (0%) | 1 (0%) | 2 (0%) |
| Screening CD4 cell count [cells/mm$^3$] | N | 1545 | 412 | 1547 | 572 | 565 | 1,137 |
| | Min-Max | 351–1,842 | 350–1,277 | 350–2033 | 351–1,836 | 353–1,787 | 351–1,836 |
| | Median (Q1-Q3) | 527 (440–652) | 543 (432–689) | 533(434–678) | 627(515–784) | 650 (514–835) | 631(514–810) |
| | 350–< 400 | 214 (14%) | 51 (12%) | 225 15%) | 23 (4%) | 29 (5%) | 52 (5%) |
| | 400–< 450 | 227 (15%) | 76 (18%) | 239 (15%) | 51 (9%) | 42 (7%) | 93 (8%) |
| | 450–< 500 | 213 (14%) | 40 (10%) | 180 (12%) | 55 (10%) | 45 (8%) | 100 (9%) |
| | 500–< 750 | 652 (42%) | 169 (41%) | 650 (42%) | 276 (48%) | 260 (46%) | 536 (47%) |
| | ≥ 750 | 239 (15%) | 76 (18%) | 253 (16%) | 167 (29%) | 189 (33%) | 356 (31%) |
| PROMISE transition | 1077BA > 1077BP | 754 (49%) | 253 (61%) | 745 (48%) | 246 (43%) | 499 (88%) | 745 (66%) |
| | 1077BA > 1077BP > 1077BM | 242 (16%) | 13 (3%) | 252 (16%) | 252 (44%) | 0 (0%) | 252 (22%) |
| | 1077BL > 1077BP | N/A | N/A | N/A | 32 (6%) | 62 (11%) | **94 (8%)** |
| | 1077BL > 1077BP > 1077BM | N/A | N/A | N/A | 34 (6%) | 0 (0%) | **34 (3%)** |
| | 1077FA > 1077BP | 6 (0%) | 0 (0%) | 9 (1%) | 5 (1%) | 4 (1%) | **9 (1%)** |
| | 1077FA > 1077BP > 1077BM | 3 (0%) | 0 (0%) | 3 (0%) | 3 (1%) | 0 (0%) | **3 (0%)** |

**Maternal disease progression and toxicity.** In the AP-only cohort, seventeen women experienced at least one confirmed primary endpoint: 11 maternal deaths (7 (64%) in the LPV/r +ZDV/3TC arm, 4 (36%) in the ZDV+sdNVP+TDF/FTC tail arm and none in the LPV/r +TDF/FTC arm. Eight AIDS defining illness (WHO Stage IV) (2 (25%) in the LPV/r +ZDV/3TC arm, 5 (63%) in the ZDV+sdNVP+TDF/FTC tail arm and 1 (12%) in the LPV/r +TDF/FTC arm. Comparing the combination ART group to ZDV+sdNVP+TDF/FTC tail, there was no statistical difference for the primary efficacy endpoint (HR = 1.14, 95%CI (0.44,

2.96), p-value = 0.79) (Fig 3A, Table 2). The primary efficacy endpoint rate was low 0.28–0.37 events per 100 person years (Table 2). There were no differences observed for secondary efficacy endpoints: Composite HIV/AIDS-related events (WHO Stage IV, pulmonary tuberculosis, and other serious bacterial infections) or death (HR = 0.94, 95% CI: (0.61, 1.45), p-value = 0.80). Likewise there were no significant differences in progression to WHO Stage Clinical II/III events (HR = 1.12, 95% CI: (0.83, 1.50), p-value = 0.45) (Table 2). Moderate, unexplained weight loss (< 10% body weight) was noted among 75 women during pregnancy; 35 (47%) received ZDV+sdNVP+TDF/FTC tail; 31 (41%) received LPV/r+ZDV/3TC while 9 (12%) received LPV/r+TDF/3TC. Only 6 women had unexplained severe weight loss (> 10% body weight) during pregnancy: 3 received ZDV+sdNVP+TDF/FTC" tail" and the other 3 received LPV/r+ZDV/3TC. None of the women receiving LPV/r+TDF/FTC had severe unexplained weight loss during pregnancy.

The incidence of grade 3–4 signs and symptoms, grade 2–4 hematological abnormalities and chemistry abnormalities was significantly higher for the LPV/r+ZDV/3TC arm compared to the ZDV +sdNVP+TDF/FTC tail arm; Hazard Ratio(HR) = 1.18, 95%CI (1.03, 1.35), p-value = 0.02) (Table 2). The difference was driven by an early separation in the survival curves through the first 24 weeks of follow-up tended to wane with the survival curves remaining similar after 108 weeks of follow-up. In period 2, no apparent by-arm differences in grade 2–4 events arm were observed: comparing LPV/r+TDF/FTC to LPV/r+ZDV/3TC (HR = 1.12, 95% CI (0.83, 1.50), p-value = 0.46), comparing LPV/r+TDF/FTC to ZDV +sdNVP+TDF/FTC tail (HR = 1.09, 95%CI (0.82, 1.45), p-value = 0.55) (Table 2). Hemoglobin adverse events were the most frequent AP-only toxicity and occurred in similar proportions across the three groups (Table 3A).

## PP- only cohort outcomes

**Starting life-long ART, changes in ART regimens and virologic failure.** Two hundred ninety six (26%) of 1129 women in the PP-only cohort, started ART for their own health or required a regimen change. Of the women who needed ART for their own health, 72% (213) had immunologic failure; and other reasons included Hepatitis B co-infection, national guidelines changes to treatment initiation, or progression to a disease of clinical WHO Stage III and TB. Five women switched their ART because of virologic failure, toxicities, or immunologic failure. By week 69 (median duration of breastfeeding), the incidence virologic failure was 21% (95% CI: (18%, 25%).

**Maternal disease progression and toxicity.** For women in the PP-only Cohort, there were no significant differences for women on ART compared to No ART, in disease progression to AIDS-defining death or death (HR = 0.76, 95% CI 0.14, 4.16, p-value = 0.75) (Fig 3B), for time to death, time to AIDS-defining illness or death, or for WHO II/III stage (Table 2). There was a marginally significant difference for composite endpoint HIV/AIDS related event or death or WHO Stage II/III (HR = 0.65, 95% CI 0.42, 1.01, p-value = 0.05). Out of six HIV/AIDS related events that occurred after 2.5 years of study follow up, four (67%) were due to tuberculosis (three events of pulmonary and one event of disseminated tuberculosis). The estimated hazard ratio for the time to the first pulmonary tuberculosis was 0.17 (95 CI 0.02, 1.34; p-value = 0.06). Moderate, unexplained weight loss (< 10% body weight) was noted among 44 women; 29 (66%) were not receiving triple ART during breastfeeding while 15 (34%) were receiving triple ART during breast feeding. Only 4 women had unexplained severe weight loss (> 10% body weight). Three (75%) were not receiving triple ART during breastfeeding while 1 (25%) was receiving triple ART during breast feeding.

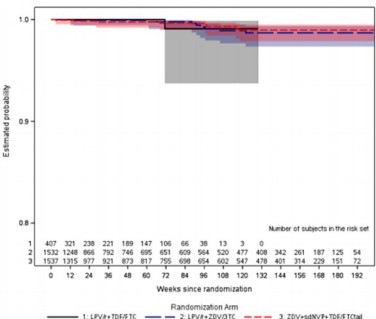
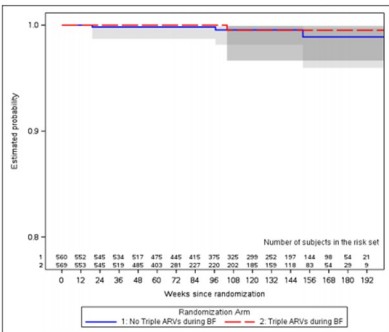

**Fig 3.** A. (AP-only): Time to first confirmed AIDS defining illness or death Fig 3B (PP-only) time to first confirmed HIV/ AIDS defining illness or death.

For the PP-only analysis group, there was no statistical difference for the composite grade 2, 3 or 4 safety endpoint (p-value = 0.95). However, in a post-hoc analysis that restricted to grade 3 and grade 4 events chemistry events, there was a significant difference between the two arms (p-value = 0.03) with the "No Triple ARVs during BF" group having a higher event rate (1.5 events per 100 person-years compared to 0.5 events per 100 person-years, Table 3B). As with the AP-only cohort, hematologic adverse events were the most frequent toxicity driven by increased neutropenia events (Table 3B).

## Discussion and conclusion

This pre-planned analysis from the PROMISE Trial allowed us to examine the impact of time-limited ART followed by treatment interruption on maternal health outcomes in two cohorts of women: the AP- only cohort who received antenatal ART compared to ZDV alone followed by no ARVs post-delivery; and the PP-only cohort, comparing women who received ART versus No ART during breastfeeding followed by no ART after end of breastfeeding. Among these two groups of women there was no statistical difference for the primary composite endpoint of AIDS defining illness/or death irrespective of whether or not the women had received short term ART during pregnancy or during breastfeeding. The AIDS/death endpoint rate was low across all PROMISE comparison groups during the 4 year follow up period. These specific cohort analyses were similar to the prior PROMISE Maternal Health findings[11] as well as the 1077HS results [8].

In the AP-only analysis, women on ZDV-based ART had a higher rate of grade 2–4 adverse events compared to ZDV +sdNVP+TDF/FTC tail. This result is likewise consistent with the previous overall PROMISE AP analysis[6] that compared adverse events by antepartum ARV regimen arm through 14 days postpartum. The observed effect in the previously published AP analysis does not appear to carry over past 14 days postpartum in these extended follow-up analyses. We did not find significant differences in progression to grade II-III events during the median 1 year of follow up for women who received antenatal ART compared to those who did not, and then received no ART following delivery. This finding suggests that if treatment interruption does occur post-delivery for women on ART during pregnancy such as might occur due to poor adherence or toxicity necessitating a temporary treatment halt, it is unlikely to result in increased risk of HIV disease progression to AIDS or death over the short term.

In the PP-only analyses, which followed up antepartum and postpartum disease progression and adverse events for women who were on ZDV +sdNVP+TRV tail during pregnancy, there

**Table 2. Summary for the primary, secondary clinical endpoints for AP-only and PP-only.**

| End point (Time to first event) | AP only (Antepartum ART/no./100py) | | | | PP only (postpartum ART/no./100py) | | |
|---|---|---|---|---|---|---|---|
| | ZDV+sdNVP +TDF/FTC | LPV/r +ZDV/3TC | LPV/r +TDF/FTC | Hazard ratio (95%CI) | Triple ARVs during BF | No Triple ARVs during BF | Hazard ratio |
| **Primary Endpoint** | | | | | | | |
| **AIDS defining illness or death** | 0.33 | 0.37 | 0.28 | 1.14 (0.44,2.96) | 0.2 | 0.32 | 0.76 (0.14,4.16) |
| **AIDS defining (WHO 4)** | 0.21 | 0.09 | 0.28 | 0.59 (0.14,2.46) | 0.1 | 0.24 | 0.48 (0.05,4.71) |
| **Death** | 0.16 | 0.33 | 0.00 | 1.84 (0.54,6.31) | 0.1 | 0.16 | 0.80 (0.07,8.78) |
| **Secondary Endpoints** | | | | | | | |
| **HIV/AIDS related events** | 1.61 | 1.52 | 1.43 | 0.89 (0.56,1.40) | 0.62 | 1.15 | 0.59 (0.22,1.54) |
| **Composite Endpoint of HIV/AIDS related event or death** | 1.73 | 1.76 | 1.43 | 0.94 (0.61,1.45) | 0.72 | 1.23 | 0.65 (0.26,1.60) |
| **Pulmonary Tuberculosis** | 0.50 | 0.28 | 0.57 | 0.67 (0.27,1.64) | 0.1 | 0.65 | 0.17 (0.02,1.34) |
| **Single Bacterial Pneumonia** | 0.58 | 0.71 | 0.28 | 1.02 (0.50,2.10) | 0.31 | 0.32 | 1.09 (0.24,4.92) |
| **Bacterial infections resulting in hospitalization** | 0.54 | 0.85 | 0.28 | 1.28 (0.63,2.59) | 0.1 | 0.08 | 1.32 (0.08,21.68) |
| **Composite Endpoint of HIV/AIDS related event and WHO stage 2 and 3 events** | 4.73 | 4.81 | 6.11 | 1.06 (0.82,1.38) | 3.29 | 5.22 | 0.65 (0.26,1.60) |
| **WHO 2 and 3 events** | 3.60 | 3.73 | 4.90 | 1.12 (0.83, 1.50) | 2.97 | 4.5 | 0.68 (0.43,1.08) |
| **Composite Endpoint of WHO 2 and 3 events or death** | 3.68 | 4.02 | 4.90 | 1.17 (0.87,1.56) | 3.07 | 4.66 | 0.69 (0.44,1.08) |
| **Composite Endpoint of WHO 2 and 3 events or AIDS defining illness or death** | 3.86 | 4.07 | 5.20 | 1.13 (0.85,1.51) | 3.07 | 4.85 | 0.66 (0.42,1.03) |
| **Toxicity Events+** | | | | | | | |
| **Grade 2, 3 and 4 Toxicity** | 22.4 | 27.7 | 33.4 | 1.18 (1.03, 1.35)[a] 1.12 (0.83, 1.50)[b] 1.09 (0.82, 1.45)[c] | 14.9 | 14.7 | 1.01 (0.79,1.28) |
| **Grade 3 and 4 Toxicity** | 9.5 | 11.5 | 14.4 | 1.16 (0.96, 1.41)[a] 1.45 (0.93, 2.25)[b] 1.35 (0.89, 2.07)[c] | 5.6 | 7.3 | 0.78 (0.55,1.11) |

[a] Comparison of LPV/r+ZDV/3TC to ZDV+sdNVP+TDF/FTC(Reference) for period 1 and period 2 of the AP-only analysis

[b] Comparison of LPV/r +TDF/FTC to LPV/r+ZDV/3TC (Reference) period 2 of the AP-only analysis

[c] Comparison of LPV/r +TDF/FTC to ZDV+sdNVP+TDF/FTC(Reference) period 2 of the AP-only analysis

were no significant differences in the primary endpoint of AIDS or death among those receiving ART compared to those not on ART. However the difference in the event rate for the composite endpoint of HIV/AIDS or death related event or WHO Stage II/III comparing the "Triple ARVs (ART) during BF" arm to the "No Triple ARVs during BF" arm was marginally significant, favoring the ART treatment arm. Likewise moderate weight loss as well as pulmonary tuberculosis was more frequent in the "No Triple ARVs during BF" arm. This

**Table 3A. Laboratory adverse events for the AP-only analysis group.**

| Toxidties | ZDC+sdNVP+TDF/FTC tall (n = 1537) Grade | | | | ZDC/3TC+LPV/r (n = 1532) Grade | | | | TDF/FTC/LPV/r (N = 3101) Grade | | | |
|---|---|---|---|---|---|---|---|---|---|---|---|---|
| Any chemistry event | 2 | 3 | 4 | Total | 2 | 3 | 4 | Total | 2 | 3 | 4 | Total |
| ALT(SGPT) | 52 (3%) | 13 (1%) | 12 (1%) | 77 (5%) | 61 (4%) | 32 (2%) | 20 (1%) | 113 (4%) | 130 (4%) | 52 (2%) | 39 (1%) | 221 (7%) |
| Creatinine | 3 (<0.5%) | 0(0%) | 4 (<0.5%) | 7 (<0.5%) | 3 (<0.5%) | 1 (<0.5%) | 1 (<0.5%) | 5 (<0.5%) | 8 (<0.5%) | 2(<0.5%) | 5 (<0.5%) | 15 (<0.5%) |
| Any hematologic event | | | | | | | | | | | | |
| Platelets | 37 (2%) | 3 (<0.5%) | 5 (<0.5%) | 45 (3%) | 39 (3%) | 7 (<0.5%) | 0(0%) | 46 (3%) | 85 (3%) | 10 (<0.5%) | 6 (<0.5%) | 101(3%) |
| Hemoglobin | 103 (7%) | 39 (3%) | 16 (1%) | 158 (10%) | 103 (7%) | 39 (3%) | 16 (1%) | 158 (10%) | 229 (7%) | 90 (3%) | 37 (1%) | 356 (11%) |
| White Blood Cells (WBC)/ Differential | 97 (6%) | 35 (2%) | 6 (<0.5%) | 138 (9%) | 87 (6%) | 35 (2%) | 6 (<0.5%) | 128 (8%) | 203 (6%) | 77 (2%) | 14 | 293 (9%) |
| Absolute Neutrophil Count | 97 | 35 | 6 | 138 | 88 | 34 | 6 | 128 | 204 | 76 | 13 | 293 |
| WBC | 4 | 1 | 0 | 5 | 3 | 2 | 0 | 5 | 7 | 4 | 0 | 11 |

finding provides some evidence of the clinical benefit of ART on moderate HIV disease progression[2].

Overall, these subgroup 1077BF/1077FF Maternal Health findings assessing the impact of short term ART during pregnancy or post-delivery during breastfeeding followed by randomized treatment interruption, are similar to the findings in the PROMISE 1077HS study conducted in non-breastfeeding settings where women received ART during pregnancy; as well as the findings in the PROMISE 1077BF/1077FF Maternal Health analyses looking at disease progression to AIDS or death following birth among women on maternal ART during pregnancy [11]. However the findings are in contrast to the SMART trial results among a large population of older adults whose episodic ART interruption significantly increased the risk of opportunistic disease or death[12].

In terms of milder disease progression and comparing women randomized to stop or continue ART post-delivery, the estimated hazard ratio for WHO Stage II/III in PROMISE 1077HS was 0.48 (95% CI (0.33, 0.70) for mothers in 1077HS; whereas in the 1077BF/1077FF maternal health analyses (11), the hazard ratio was 0.60 (0.39, 0.90), similar to the hazard ratio of 0.69 (0.44, 1.08) reported in these analyses.

The incidence of virologic failure at 24 weeks in the AP only cohort was 7% while that in the PP only cohort was 21% at 69 weeks. These data reflect high rates of virologic failure during the postpartum period and this is consistent with other literature demonstrating poor maternal

**Table 3B. Laboratory adverse events for the PP-only analysis group.**

| Toxidties | Triple ARVs during BF (N = 569) Grade | | | | No triple ARVs during BF (N = 560) Grade | | | |
|---|---|---|---|---|---|---|---|---|
| Any chemistry event | 2 | 3 | 4 | Total | 2 | 3 | 4 | Total |
| ALT(SGPT) | 14 (2%) | 5 (1%) | 0 (0%) | 19 (3%) | 15 (3%) | 8 (1%) | 10 (2%) | 33 (6%) |
| Creatinine | 1(<0.5%) | 0(0%) | 0(0%) | 1 (<0.5%) | 0(<0%) | 0(<0%) | 2(<0.5%) | 2(<0.5%) |
| Any hematologic event | | | | | | | | |
| Platelets | 5 (1%) | 2(<0.5%) | 0(0%) | 7 (1%) | 17 (3%) | 2(<0.5%) | 5(1%) | 24 (4%) |
| Hemoglobin | 8 (1%) | 2(<0.5%) | 3 (1%) | 13 (2%) | 14 (3%) | 7 (1%) | 3 (1%) | 24 (4%) |
| White Blood Cells (WBC)/Differential | 58 (10%) | 15 (3%) | 5(1%) | 78 (14%) | 47 (8%) | 19 (3%) | 2(<0.5%) | 68 (12%) |
| Absolute Neutrophil Count | 58 | 15 | 5 | 78 | 47 | 19 | 2 | 68 |
| WBC | 2 | 0 | 0 | 2 | 2 | 0 | 0 | 2 |

ART drug adherence post-delivery both in resource rich as well as resource limited settings [4, 13].

This study further contributes to our understanding of the impact of ART on HIV disease progression among pregnant and breastfeeding post-partum HIV infected women living in resource limited settings by analyzing the effects of short term ART, followed by treatment interruption. The analyses highlight a number of important if unexpected findings regarding the general good health of HIV infected mothers in resource limited international settings, who were primarily young and asymptomatic at entry into PROMISE trial: First, it demonstrates a relatively low rate of disease progression to HIV/AIDS, among HIV infected pregnant African mothers, irrespective of whether the women were on ZDV or an ART regimen antepartum. Second, interruption of ART after pregnancy, based on randomization to infant NVP prophylaxis during breastfeeding, did not result in significantly increased rates of disease progression that would have been expected based on earlier data from the treatment discontinuation arm of the SMART trial whose follow up period was shorter[12]. Similarly, not being on triple ART as in the START trial, did not result in increased rates of disease progression. This is likely due to the younger age, high CD4 counts of the PROMISE population and the lack of prior immunodeficiency compared to SMART and START participants. It is important to note that while SMART trial involved interruption of triple ART treatment, this study's treatment interruption was of daily zidovudine during pregnancy followed by a TDF/FTC "tail" at labor and delivery. Whereas continuation of lifetime ART without interruption is recommended by WHO as standard of care from the time of diagnosis and has the added benefit of reducing the risk of transmission to partners among those who do achieve undetectable viral loads, these findings provide some reassurance that there was no evidence of increased risk of AIDS/death over 1–4 years of follow up for women who may face challenges adhering to ART continuously during the post-partum period, or in situations, where treatment must be temporarily interrupted due to toxicity or drug intolerance.

There are a number of strengths to these PROMISE trial Maternal Health sub cohort analyses. First, the findings are based on robust clinical trial data in pre-specified comparisons that were gathered on pregnant and post-partum HIV infected women from multiple international sites; which increases generalizability of the findings. Second, the trial sample size is large with excellent follow up and retention; and third, the findings of lack of HIV progression are consistent with those from the 1077HS study done in middle level and high level income settings [8].

Limitations of the study include the lower than expected rates of disease progression than anticipated in PROMISE, limiting our statistical power to observe a difference in rates of disease progression. In addition, the findings on higher rates of laboratory toxicity with ART compared to ZDV alone relate to the specific regimens used in PROMISE (lopinavir/ritonavir and Zidovudine) and may not be generalizable to other ART regimens.

In conclusion, in this analyses of the PROMISE 1077BF/1077FF trial, we found low rates and no significant differences for disease progression to AIDS or death among young asymptomatic HIV infected pregnant and post-partum breastfeeding women exposed to short term ART either during pregnancy or breastfeeding; followed by randomized observation discontinuing ART during up to 4 years follow-up in the trial. Given the global programmatic challenges in maintaining maternal adherence to taking ART particularly post-delivery in programmatic settings, these data suggest that short term lapses in treatment do not result in more rapid disease progression to AIDS or death in this young age group of pregnant and postpartum HIV infected mothers. However continued efforts to identify simpler better tolerated high adherence treatment regimens for this population remains a priority.

## Supporting information

**S1 Data.**
(ZIP)

**S1 File. Sensitivity analyses (Pre-specified log-rank test vs post-hoc permutation).**
(DOCX)

**S1 Consort checklist. CONSORT 2010 checklist of information to include when reporting a randomised trial.**
(DOC)

## Acknowledgments

We gratefully acknowledge the contributions of the study staff, site investigators and site staff who conducted IMPAACT 1077BFstudy:

**PROMISE Study Team Members**: Judith Currier, Katherine Luzuriaga, Adriana Weinberg, James McIntyre, Tsungai Chipato, Karin Klingman, Renee Browning, Mireille Mpoudi-Ngole, Jennifer S. Read, George Siberry, Heather Watts, Lynette Purdue, Terrence Fenton, Linda Barlow-Mosha, Mary Pat Toye, Mark Mirochnick, William B. Kabat, Benjamin Chi, Marc Lallemant, Karin Nielsen; **Statistical and Data Analysis Center**, Harvard T.H. Chan School of Public Health: Kevin Butler MS, Konstantia Angelidou MS, David Shapiro PHD, and Sean Brummel Ph.D. **IMPAACT Operations Center**: Anne Coletti, Veronica Toone, Megan Valentine, Kathleen George; **Frontier Science Data Management Center**: Amanda Zadzilka, Michael Basar, Amy Jennings, Adam Manzella.

**INDIA**. Sandesh Patil, MBBS; Ramesh Bhosale, MD; Neetal Nevrekar, MD **MALAWI. Blantyre**: Salome Kunje, BSc; Alex Siyasiya, Certificate in Microbiology; Mervis Maulidi, Certificate in Nursing and Midwifery **Lilongwe/UNC**: Francis Martinson, MBChB; Ezylia Makina, RNM; Beteniko Milala, BAE **SOUTH AFRICA. Durban Paediatric**: Nozibusiso Rejoice Skosana, BN; Sajeeda Mawlana, MBChB **Family Clinical Research Unit**: Jeanne Louw MSc; Magdel Rossouw MNutr, MBChB; Lindie Rossouw MBChB. **Shandukani Research**: Masebole Masenya, MD; Janet Grab, BPharm **Soweto**: Nasreen Abrahams, MBA; Mandisa Nyati, MBChB; Sylvia Dittmer, MBChB;**Umulazi CRS** Dhayendre Moodley, MSc, PhD; Vani Chetty, BScHon; Alicia Catherine Desmond, MPharm **TANZANIA. Kilimanjaro Christian Medical Centre** Boniface Njau, MPH; Cynthia Asiyo, Bsc; Pendo Mlay, MD; **UGANDA. MU-JHU Research Collaboration**: Maxensia Owor MBChB, Mmed, MPH, Moreen Kamateeka, MBChB, MPH; Dorothy Sebikari MBChB, MPH; **ZAMBIA. George Clinic**: Felistas M. Mbewe, RN, BSc; Martin Mwalukanga, Diploma in Clinical Medicine **ZIMBABWE. Harare Family Care**: Tichaona Vhembo, MBChB; Nyasha Mufukari, Bpharm **Seke North**: Lynda Stranix-Chibanda, MBChB; Teacler Nematadzira, MBChB; Gift Chareka, MSc. **St. Mary's**: Jean Dimairo, Bpharm, Tsungai Chipato MB ChB FRCOG; Bangani Kusakara MBChB; Mercy Mutambanengwe BPharm; Emmie Marote SRN MA.

## Author Contributions

**Conceptualization:** Judith S. Currier, Mary Glenn Fowler.

**Data curation:** Sean S. Brummel.

**Formal analysis:** Sean S. Brummel, Konstantia Angelidou.

**Funding acquisition:** Judith S. Currier, Mary Glenn Fowler.

**Investigation:** Patience Atuhaire, Judith S. Currier, Mary Glenn Fowler.

**Methodology:** Judith S. Currier, Mary Glenn Fowler.

**Project administration:** Judith S. Currier, Mary Glenn Fowler.

**Software:** Sean S. Brummel.

**Supervision:** Patience Atuhaire, Sean S. Brummel, Blandina Theophil Mmbaga, Lee Fairlie, Avy Violari, Gerhard Theron, Cornelius Mukuzunga, Sajeeda Mawlana, Mwangelwa Mubiana-Mbewe, Megeshinee Naidoo, Bonus Makanani, Patricia Mandima, Teacler Nematadzira, Nishi Suryavanshi, Tapiwa Mbengeranwa, Amy Loftis, Michael Basar, Katie McCarthy, Judith S. Currier, Mary Glenn Fowler.

**Validation:** Sean S. Brummel, Konstantia Angelidou.

**Visualization:** Sean S. Brummel, Konstantia Angelidou, Judith S. Currier, Mary Glenn Fowler.

**Writing – original draft:** Patience Atuhaire, Mary Glenn Fowler.

**Writing – review & editing:** Patience Atuhaire, Sean S. Brummel, Blandina Theophil Mmbaga, Konstantia Angelidou, Lee Fairlie, Avy Violari, Gerhard Theron, Cornelius Mukuzunga, Sajeeda Mawlana, Mwangelwa Mubiana-Mbewe, Megeshinee Naidoo, Bonus Makanani, Patricia Mandima, Teacler Nematadzira, Nishi Suryavanshi, Tapiwa Mbengeranwa, Amy Loftis, Michael Basar, Katie McCarthy, Judith S. Currier, Mary Glenn Fowler.

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
