## [Decision Letter · Decision Letter 0]

24 Oct 2019

PONE-D-19-25671

The impact of short term Antiretroviral Therapy (ART) interruptions on longer term maternal health outcomes – a randomized clinical trial

PLOS ONE

Dear Dr Atuhaire,

Thank you for submitting your manuscript to PLOS ONE. After careful consideration, we feel that it has merit but does not fully meet PLOS ONE’s publication criteria as it currently stands. Therefore, we invite you to submit a revised version of the manuscript that addresses the points raised during the review process. 

You will see that the Referees found your work of some interest. However, they also raised major criticisms and did not grant your paper enough priority to recommend publication. However, if you think all objections raised by the referees can be considered and if additional data requested by reviewers can be provided, we may be willing to reconsider your manuscript. Please respond to all the comments by Reviewers #1, # 2 and #3 with special attention to the concern raised by Reviewer #1.

We would appreciate receiving your revised manuscript by the next 12 weeks. To enhance the reproducibility of your results, we recommend that if applicable you deposit your laboratory protocols in protocols.io, where a protocol can be assigned its own identifier (DOI) such that it can be cited independently in the future. For instructions see: http://journals.plos.org/plosone/s/submission-guidelines#loc-laboratory-protocols

We look forward to receiving your revised manuscript.

Kind regards,

Giuseppe Vittorio De Socio, MD, PhD

Academic Editor

PLOS ONE

**Journal Requirements:**

2.  Thank you for including your ethics statement:  "All women provided written informed consent. The study was approved by local and collaborating institutional review boards and other relevant regulatory authorities; and was reviewed for safety and efficacy by an independent Data and Safety Monitoring Board (DSMB).".   

a.Please amend your current ethics statement to include the full name of the ethics committee/institutional review board(s) that approved your specific study.

b.Once you have amended this/these statement(s) in the Methods section of the manuscript, please add the same text to the “Ethics Statement” field of the submission form (via “Edit Submission”).

**Comments to the Author**

1. Is the manuscript technically sound, and do the data support the conclusions?

Reviewer #1: Yes

Reviewer #2: Partly

Reviewer #3: Yes

2. Has the statistical analysis been performed appropriately and rigorously? 

Reviewer #1: Yes

Reviewer #2: Yes

Reviewer #3: Yes

3. Have the authors made all data underlying the findings in their manuscript fully available?

Reviewer #1: Yes

Reviewer #2: Yes

Reviewer #3: Yes

4. Is the manuscript presented in an intelligible fashion and written in standard English?

Reviewer #1: Yes

Reviewer #2: No

Reviewer #3: Yes

5. Review Comments to the Author

Reviewer #1: This pre-planned analysis from PROMISE trial evaluated the impact of short term antretroviral therapy, followed by periods of treatment interruption on long- term maternal health outcomes.

Specifically this analysis focused on two pre-specified comparisons of the effect of maternal combination ART used only during the ante-partum period and the effect of maternal combination ART given only during the breastfeeding period followed by treatment interruption unless the women met standard -of –care treatment criteria.

The primary outcome was progression to AIDS or death. Secondary efficacy outcomes included time to WHOII/III clinical events. The secondary safety outcome included selected Grade 2 laboratory abnormalities and all Grade 3 or higher laboratory values and signs and symptoms.

The results showed that the rate of progression to AIDS and/or death was similar and low across all the comparison arms.

So Authors underline that this result provides reassurance that there were limited consequences for short term ART, followed by treatment interruptions, among pregnant and breastfeeding post -partum asymptomatic HIV infected women living in resource limited settings.

The study procedures are correct, sample size adequate, study outcomes clear, statistical analysis done well and the whole contribution is relevant.

I would like to ask Authors just a few questions:

Can they better clarify what they mean with treatment interruption (duration, start and stop criteria)?

Can they comment on the apparent contrast with the 2015 WHO recommendation to use lifelong ART initiated at the time of diagnosis?

Bearing in mind that people with undetectable viral load are significantly less likely to transmit virus and collectively individuals with lower viral load lead to communities with lower viral load so with less risk of HIV transmission, can Authors comment on that?

In order to ameliorate adherence even in resource limited settings, would Authors suggest to test simpler and better tollerated regimens instead of shorter periods of treatment?

Reviewer #2: Authors presented a sub-study of a trial conducted from 2011 to 2014 in Africa and India.

I found this study of limited interest in its field. The first reason is that the study has been conducted from 2011 to 2014 and I feel that these data are too much old to be applied today in clinical settings. International guidelines recommendations are constantly evolving; therefore, we have now robust evidence that immediate ART start is important to have a good clinical outcome.

The observation that in 1-4 years we will not have an excess in mortality is related to the fact that people included in this trial were almost immunocompetent. However, this explanation is not sufficient to understand these results. Moreover, the message that ART could be delayed I feel is quite dangerous for scientific community.

The comparison with START trial is not appropriate in this point of view. Nonetheless, the strong recommendations to have universal access to ART is related not only to the mortality risk but also to reach the goal in reducing chronic inflammation and immune activation.

I found the study almost forced in the conclusions.

Abstract: in the abstract are unclear the treatment options and the outcomes. Authors stated “The first analysis compared ART use limited to the antepartum period (AP-only) relative to women randomized to Zidovudine”. This is not an objective of a RCT.

The objectives and Methods are unclear and in particular is not clear as the RCT PROMISE is involved in this study. Actually, in the result section, Authors seem to compare two cohort and not two arms of a RCT.

Moreover, the paper is difficult to understand: study procedures are confusing and nor clear. Discussion is too long

Reviewer #3: This is a model of a well-written clinical trials paper from a statistical standpoint, following the CONSORT document to the letter. I have 2 minor questions:

1. In the original sample size section on page 10, it would be helpful to know what was observed, as well as what was assumed.

2. The lower than expected rates of disease progression are problematic. Was it considered to analyze the data using

techniques for small event rates (e.g., exact analyses) given that we have the computational resources now to do rare events simulation and exact nonparametric inference. Even more interesting would be a randomization test using censored logrank scores and re-randomization of 100,000 randomization sequences (see Rosenberger and Lachin, 2016, Randomization in Clinical Trials, Wiley).

6. PLOS authors have the option to publish the peer review history of their article (what does this mean?). If published, this will include your full peer review and any attached files.

Reviewer #1: No

Reviewer #2: No

Reviewer #3: No

---

## [Author Response · Author response to Decision Letter 0]

21 Dec 2019

Comments from reviewer 1

This pre-planned analysis from PROMISE trial evaluated the impact of short term antretroviral therapy, followed by periods of treatment interruption on long- term maternal health outcomes.

Specifically this analysis focused on two pre-specified comparisons of the effect of maternal combination ART used only during the ante-partum period and the effect of maternal combination ART given only during the breastfeeding period followed by treatment interruption unless the women met standard -of –care treatment criteria.

The primary outcome was progression to AIDS or death. Secondary efficacy outcomes included time to WHOII/III clinical events. The secondary safety outcome included selected Grade 2 laboratory abnormalities and all Grade 3 or higher laboratory values and signs and symptoms.

The results showed that the rate of progression to AIDS and/or death was similar and low across all the comparison arms.

So Authors underline that this result provides reassurance that there were limited consequences for short term ART, followed by treatment interruptions, among pregnant and breastfeeding post -partum asymptomatic HIV infected women living in resource limited settings.

The study procedures are correct, sample size adequate, study outcomes clear, statistical analysis done well and the whole contribution is relevant.

Comment: I would like to ask Authors just a few questions:

Can they better clarify what they mean with treatment interruption (duration, start and stop criteria)?

Response: Thank you for this comment. Treatment Interruption in PROMISE 1077BF/1077FF for this analyses refers to the stopping of antepartum ART at the time of randomization to the next component of PROMISE; and in the postpartum breastfeeding component refers to the randomized stopping of ART at the end of breastfeeding for the subgroup of women randomized to ART during breastfeeding. We have reworded to clarify the duration, start and stop criteria of the triple ARV regimen in the antepartum period. This is highlighted in lines 139-146.

Comment: Can they comment on the apparent contrast with the 2015 WHO recommendation to use lifelong ART initiated at the time of diagnosis? Bearing in mind that people with undetectable viral load are significantly less likely to transmit virus and collectively individuals with lower viral load lead to communities with lower viral load so with less risk of HIV transmission, can Authors comment on that?

Response: Thank you for raising this. PROMISE was designed and implemented prior to the 2015 WHO guidelines and part of its overall goal was to provide clinical trial data among pregnant and postpartum women to help inform the WHO guidelines. During the time PROMISE was being implemented there were several rapidly evolving WHO guidelines around prevention, treatment and breastfeeding recommendations. The PROMISE team worked to respond to these evolving guidelines with amendments as well as consultations with the MoH as to uptake of guidelines and timing. As noted in the Statistical analyses section (lines 260-263), when the START results came out in mid June 2015, the PROMISE team immediately reviewed the results; and in July 7, 2015, ended randomization in PROMISE and recommended initiation of ART to all PROMISE mothers. Thus the data in these analyses are only through the period prior to ending the randomized trial and the WHO updated guidelines. We definitely agree that the short term treatment interruptions resulting from the PROMISE sequential randomizations, while in keeping with ART and PMTCT guidelines at the time, are not the current WHO “Test and Treat” recommendations for initiation of lifetime ART at the time of diagnosis; and which has the added benefit of reducing risk of transmission to partners if the person achieves non-detectable viral load. In the discussion, we have added a sentence re current WHO guidelines (lines 442-444) to remind readers of the benefits of undetectable viral load in terms of reducing the risk of transmission to partners. 

Comment: In order to ameliorate adherence even in resource limited settings, would Authors suggest to test simpler and better tolerated regimens instead of shorter periods of treatment?

Response: Thank you for this comment and yes we agree. This recommendation now is stated in the last sentence of the manuscript lines 470-471.

Comments from reviewer 2

Comment: Authors presented a sub-study of a trial conducted from 2011 to 2014 in Africa and India.

I found this study of limited interest in its field. The first reason is that the study has been conducted from 2011 to 2014 and I feel that these data are too much old to be applied today in clinical settings. International guidelines recommendations are constantly evolving; therefore, we have now robust evidence that immediate ART start is important to have a good clinical outcome. The observation that in 1-4 years we will not have an excess in mortality is related to the fact that people included in this trial were almost immunocompetent. However, this explanation is not sufficient to understand these results. Moreover, the message that ART could be delayed I feel is quite dangerous for scientific community. The comparison with START trial is not appropriate in this point of view. 

Response: We thank you for this comment. Despite this data having been obtained in 2011 - 2015; we feel that since this is clinical trial data that answers a question as to the short term effects of going off ART among a population of pregnant and postpartum mothers who in the real world have well documented risk periods for non-adherence including in the immediate postpartum period as well as when the period of risk of transmission to their infant is over (e.g. at the end of breastfeeding). Given this reality and continued challenges with poor adherence for many mothers, we believe these findings remain highly relevant in the current programmatic settings where poor adherence remains a significant challenge. This clinical trial data therefore gives us data on the risk of disease progression or short treatment interruption among women with higher CD4 counts, who may experience adherence challenges or who do not tolerate their prescribed ART due to toxicities.. 

Comment: Nonetheless, the strong recommendations to have universal access to ART is related not only to the mortality risk but also to reach the goal in reducing chronic inflammation and immune activation. I found the study almost forced in the conclusions.

Response: Thank you for your comments. This study’s primary outcome was to determine progression to AIDS and/or death. This subset analyses did not assess immune biomarkers but that analyses is planned in a larger analyses of the mothers followed in PROMISE using stored samples to assess chronic inflammation and immune activation.

Comment: Abstract: in the abstract are unclear the treatment options and the outcomes. Authors stated “The first analysis compared ART use limited to the antepartum period (AP-only) relative to women randomized to Zidovudine”. This is not an objective of a RCT.

The objectives and Methods are unclear and in particular is not clear as the RCT PROMISE is involved in this study. Actually, in the result section, Authors seem to compare two cohort and not two arms of a RCT. Moreover, the paper is difficult to understand: study procedures are confusing and nor clear. Discussion is too long

Response: Thank you for this comment. There are two cohorts of women for each with its randomization arms. Figure 1 illustrates the composition of the two cohorts and the time of follow up. 

Comments from reviewer 3

Comment: This is a model of a well-written clinical trials paper from a statistical standpoint, following the CONSORT document to the letter. I have 2 minor questions:

1. In the original sample size section on page 10, it would be helpful to know what was observed, as well as what was assumed.

Response: Thank you for raising this. We have included the observed numbers per cohort on page 10 lines 256-258

2. The lower than expected rates of disease progression are problematic. Was it considered to analyze the data using techniques for small event rates (e.g., exact analyses) given that we have the computational resources now to do rare events simulation and exact nonparametric inference. Even more interesting would be a randomization test using censored logrank scores and re-randomization of 100,000 randomization sequences (see Rosenberger and Lachin, 2016, Randomization in Clinical Trials, Wiley).

Response: Thank you for this comment. We did not test group differences with additional modeling strategies. We believe that the event rates are so low with such small differences that the point estimates for the comparisons are not clinically relevant. Therefore, we do not think that a statistical difference would add to the study conclusion. In addition, we thought it was important to use the pre-specified testing procedure. However, we agree that with such a low event rate assuming an asymptotic distribution for testing is problematic. We took the reviewers advice and conducted a permutation test that appropriately accounted for the block randomizations and the stratification factors. We conducted 100,000 permutations resulting in 100,000 realizations of the log-rank test statistic from the null distribution. The permutation then results in a p-value of 0.19 for the antepartum comparison and a p-value of 0.31 for the breastfeeding comparison. This is compared to a p-value of 0.79 for the antepartum comparison and a p-value of 0.75 for the postpartum comparison. The study conclusions remain the same based on the permutation test. A table comparing these results was added under supporting information (supplemental table 3).

---

## [Editor Report · Decision Letter 1]

7 Jan 2020

The impact of short term Antiretroviral Therapy (ART) interruptions on longer term maternal health outcomes – a randomized clinical trial

PONE-D-19-25671R1

Dear Dr. Atuhaire,

We are pleased to inform you that your manuscript has been judged scientifically suitable for publication and will be formally accepted for publication once it complies with all outstanding technical requirements.

With kind regards,

Giuseppe Vittorio De Socio, MD, PhD

Academic Editor

PLOS ONE
---

## [Editor Report · Acceptance letter]

10 Jan 2020

PONE-D-19-25671R1 

The impact of short term Antiretroviral Therapy (ART) interruptions on longer term maternal health outcomes – a randomized clinical trial 

Dear Dr. Atuhaire:

I am pleased to inform you that your manuscript has been deemed suitable for publication in PLOS ONE. Congratulations! Your manuscript is now with our production department. 

With kind regards,

on behalf of

Dr. Giuseppe Vittorio De Socio 

Academic Editor

PLOS ONE